# CD33 Expression and Gentuzumab Ozogamicin in Acute Myeloid Leukemia: Two Sides of the Same Coin

**DOI:** 10.3390/cancers13133214

**Published:** 2021-06-28

**Authors:** Matteo Molica, Salvatore Perrone, Carla Mazzone, Pasquale Niscola, Laura Cesini, Elisabetta Abruzzese, Paolo de Fabritiis

**Affiliations:** 1Hematology Unit, S. Eugenio Hospital, ASL Roma 2, 00144 Rome, Italy; cmazzone1@virgilio.it (C.M.); pasquale.niscola@aslroma2.it (P.N.); laura.cesini@aslroma2.it (L.C.); elisabetta.abruzzese@uniroma2.it (E.A.); paolo.de.fabritiis@uniroma2.it (P.d.F.); 2Hematology, Polo Universitario Pontino, S.M. Goretti Hospital, 04100 Latina, Italy; S.perrone@ausl.latina.it; 3Department of Biomedicina and Prevenzione, Tor Vergata University, 00133 Rome, Italy

**Keywords:** acute myeloid leukemia, gemtuzumab ozogamicin, CD33, chemotherapy, sinusoidal obstructive syndrome (SOS)

## Abstract

**Simple Summary:**

Roughly 85–90% of adult and pediatric acute myeloid leukemia (AML) are CD33-positive. Gemtuzumab ozogamicin (GO), a humanized murine IgG4 anti-CD33 antibody, is the first target therapy approved in AML therapeutic scenario. This review focuses on current biological information and clinical data from several studies investigating the use of GO in patients with AML. Over the years, flow cytometry, cytogenetics, molecular techniques, and genotyping studies of CD33 SNPs have provided a comprehensive analysis of promising biomarkers for GO responses and have potentially helped to identify subgroups of patients that may benefit from GO addition to standard chemotherapies. Increased understanding of molecular mutations, altered intracellular pathways, and their potential relationship with CD33 expression may open new therapeutic landscapes based on combinatorial regimens in an AML scenario.

**Abstract:**

Acute myeloid leukemia (AML), the most frequent acute leukemia in adults, has been historically treated with infusional cytarabine (ara-c) + daunorubicin (3 + 7) for at least 40 years. The first “target therapy” to be introduced was the monoclonal anti-CD33 gemtuzumab ozogamicin (GO) in 2004. Unfortunately, in 2010 it was voluntarily withdrawn from the market both for safety reasons related to potential liver toxicity and veno-occlusive disease (VOD) and because clinical studies failed to confirm the clinical benefit during induction and maintenance. Seven years later, GO was re-approved based on new data, including insights into its mechanism of action on its target receptor CD33 expressed on myeloid cells. The present review focuses on current biological information and clinical data from several studies investigating GO. Cytogenetic, molecular, and immunophenotypic data are now able to predict the potential positive advantages of GO, with the exception of high-risk AML patients who do not seem to benefit. GO can be considered a ‘repurposed drug’ that could be beneficial for some patients with AML, mostly in combination with new drugs already approved or currently in testing.

## 1. CD33 Expression in Normal Myelopoiesis

CD33, located on chromosome 19q13.3, is a sialic acid-binding immunoglobulin-related lectin (siglec) that functions as a transmembrane receptor on hematopoietic cells [1,2]. CD33 is comprised of an amino-terminal variable (V)-set Ig-like domain mediating sialic acid binding, a C2-set Ig-like domain in its extracellular region, a transmembrane domain, and a cytoplasmic tail that contains two conserved tyrosine-based inhibitory signaling motifs (ITIM), a property shared with all CD33-related siglecs discovered to date [3]. Alternative splicing of the CD33 RNA leads to a shorter isoform expressed on the cell surface. This isoform lacks the V-set Ig-like domain as well as the disulfide bond linking the V-and C2-set Ig-like domains [3]. It preferentially binds to α2-6-and α2-3-sialylated glycans and strongly binds to sialylated ligands on leukemic cell lines [4,5]. Downstream steps in CD33 signaling are not well known, but some experimental models report the involvement of several pathways, such as Syk, c-Cbl, Vav, and ZAP-70 [6,7]. CD33 works as an inhibitory receptor; when phosphorylated on its cytoplasmic immunoreceptor tyrosine-based inhibitory motif (ITIM) sequence, it recruits SHP-1 and SHP-2 phosphatases and downregulates cell activation, in both myeloid cell lines and activated natural killer (NK) cells [8].

Increasing evidence suggests that CD33 with inhibitory signaling motifs modulates inflammatory and immune responses through the dampening of tyrosine kinase-driven signaling pathways. In vitro studies have showed that CD33 constitutively inhibits the production of pro-inflammatory cytokines such as TNF-α, IL-1β, and IL-8 by human monocytes in a sialic acid ligand-dependent and SOCS3-dependent manner [6]. Conversely, reduction of cell surface CD33 expression or interruption of sialic acid binding can augment p38 mitogen-activated protein kinase (MAPK) activity and enhance cytokine secretion, as well as cytokine-induced cellular proliferation [6].

Physiologically, CD33 expression is restricted to early multi-lineage hematopoietic progenitors (CFU-GEMM, CFU-GM, CFU-G, and E-BFU), myeloblasts and monoblasts, monocytes/macrophages, granulocyte precursors (with decreasing expression with maturation), and mast cells. On CD34+/CD33+ bone marrow cells, CD33 expression has been estimated to average around 8 × 10^3^ molecules/cell, although levels vary widely (1–20 × 10^3^ molecules/cell) [9]. Mature granulocytes may show a very low level of CD33 expression. In contrast, CD33 is not expressed outside the hematopoietic system nor is it expressed in pluripotent hematopoietic stem cells, erythrocytes, platelets, B, T, or NK cells [9].

## 2. CD33 Expression in Acute Myeloid Leukemia

The CD13 and CD33 antigens and myeloperoxidase that characterize normal myeloid progenitors are abundantly expressed on almost all acute myeloid leukemia (AML) cells [10]. Roughly, 85–90% of adult and pediatric AML are CD33-positive, as defined by the expression of this specific antigen on 20–25% or more of leukemic blasts. CD33 expression is essentially universal in acute promyelocytic leukemia (APL), with cells typically surrounded by large amounts (95–100%) of the antigen [11]. In addition, CD33 is also found on abnormal cells of other myeloid neoplasms (e.g., myelodysplastic syndromes and myeloproliferative neoplasms) [12], and on subsets of B-cell and T-cell acute lymphoblastic leukemia (ALL)/lymphoblastic lymphomas [13,14] perhaps consistent with its occasional expression on normal non-myeloid cells.

Bone marrow blasts of AML patients express between 709 and 54,894 CD33 molecules/cell (mean 10,380 molecules/cell), compared to only 859–5137 molecules/cell (mean 2997 molecules/cell) detected in normal CD33-positive bone marrow cells [12]. The CD33+/CD34+ AML subpopulations also express higher levels of CD33 antigen (mean 9482 molecules/cell) than normal bone marrow CD33+/CD34+ counterparts (mean 8154 molecules/cell) [12]. Furthermore, lower levels of CD33 have been detected on CD33+/CD34+ leukemic blasts (mean 7607 molecules/cell) of peripheral blood compared with bone marrow blasts, although most studies have not measured antigen levels on bone marrow and peripheral blood populations in individual patients [12]. CD33 expression in AML blasts using multi-parameter flow cytometry (MFC) is routinely measured both by median fluorescence intensity (CD33-MFI) and as a percentage of CD33 positivity. CD33-MFI is often detected in immunophenotypically immature CD34+CD38low stem/progenitor cells (SPC) [15]. A broad range of CD33-MFI and %CD33-positivity values are generally observed; therefore, in several studies, patients were grouped into increasing quartiles (Q1, Q2, Q3, Q4) according to CD33 blast expression levels [16,17,18].

Several studies have evaluated the correlation between CD33 expression and disease characteristics in AML patients [16,17,18]. In both adult and children, *FLT3-ITD* and *NPM1* mutations were found to be significantly heightened in prevalence with increasing CD33 expression [13,16]. Moreover, intermediate-risk AML patients lacking these mutations were inversely associated with CD33 expression. In contrast, the significant increase in the prevalence of the protein CCAAT/enhancer-binding protein alpha (CEBPA) with increasing CD33 expression was not observed when analyzed by quartiles [16]. Core binding factor (CBF)-AML (t(8;21) (q22;q22) or inv(16) (p13q22)/t(16;16) (p13;q22) was also found to be inversely correlated with CD33 expression [16,18]. Khan et al. interestingly showed that the CD34+CD38- fraction of adult patients with CBF-AML expressed CD33, while patients with intermediate- and adverse-risk disease displayed a more heterogeneous CD34+CD38- fraction, containing significant numbers of CD33-negative cells [18]. This suggests that CBF-AML may have different patterns of CD33 expression, and removal of CD33+ leukemic stem cells (LSCs) may explain the favorable clinical response seen in adult patients receiving anti-CD33 therapy. Analysis from 1583 patients in the UK-NCRI-AML17 (younger adults) and UK-NCRI-AML16 (older adults) trials showed that cytogenetic adverse risk was associated with lower CD33 expression, while intermediate-risk cytogenetics significantly increased in prevalence with increasing CD33 quartiles [16]. In the pediatric AML population, there was an inverse association between CD33 expression and prevalence of low-risk AML; in contrast, the prevalence of standard-risk disease increased significantly with increasing quartile. There was no statistically significant trend in prevalence by quartile for high-risk disease [17]. Furthermore, pediatric trials showed that patients whose AML blasts display high CD33 levels experienced inferior disease-free periods and overall survival when treated with conventional chemotherapy that did not include CD33-targeted agents [17,19].

In AML patients, CD33 can also be detected as soluble protein in the circulation and may provide some prognostic information [20], however, its role as a predictive biomarker is still controversial. It is also unclear whether soluble CD33 might interfere with the therapeutic efficacy of CD33 antibodies, although some in vitro evidence suggests that soluble CD33 may not affect the activity of CD33-targeted immunotherapy [21].

## 3. Gemtuzumab Ozogamicin (GO) and Mechanism of Action

Gemtuzumab ozogamicin (GO) is a humanized murine IgG4 anti-CD33 antibody (P67.6) conjugated with a calicheamicin hydrazide derivative attached to the oxidized carbohydrates [22]. Unfortunately, 50% of the antibody remains unconjugated and the bare antibodies, like lintuzumab, retain limited clinical activity [23], yet, in theory, could deplete available CD33 on AML blasts without exerting cytotoxicity. CD33 is continually re-expressed on myeloid cells and returns to normal expression levels within 72 h from first GO infusion [24,25]. These data resulted in the advent of a fractioned schedule (days 1, 4, 7) at lower doses (3 mg/m^2^) than the first studies, which had the advantage of limiting hepatic toxicity (sinusoidal obstruction syndrome SOS or VOD). Mechanisms of SOS are not completely elucidated and are shared with other calicheamicin-bound antibody–drug conjugates (ADC) like inotuzumab ozogamicin (anti-CD22 antibody, approved for ALL) and vadastuximab talirine (SGN-CD33A with pyrrolobenzodiazepine as a different payload) [26]. Moreover, the parent antibody (P67.6) used for GO recognizes the V-set domain of CD33 and therefore interacts with CD33^FL^ and CD33^E7a^ but not with CD33^ΔE2^, nor with CD33^ΔE2, E7a^. In fact, a SNP placed in the splice enhancer region of the CD33 gene exon 2, rs12459419 (C > T; Ala14Val) is responsible for the lack of the V-set domain of CD33 and could impair the interaction of the drug with its receptor [27].

After GO binds to surface of CD33, the complex is internalized and transferred to the lysosomes where the acidic environment causes hydrolysis of the linker and the release of calicheamicin [28] (Figure 1). Nevertheless, efflux pump activity present in a large proportion of AML cells as the multidrug resistance-associated protein 1 (ABCC1 or MRP1) and MDR1 (or P-glycoprotein or ABCB1) can limit the effect of the free drug and accounts for the poor clinical outcome observed in studies [29]. This is also the case for GO, where higher expression of MDR1 is associated with poor outcome [30,31,32], and CD33 expression is inversely correlated with MDR1 drug efflux activity [3].

Calicheamicin is a natural antibiotic of the nine-membered enediyne core family, with an anti-cancer effect 4000-fold more potent than adriamycin [33]. Calicheamicin produces a benzenoid diradical that, when positioned within the minor groove of DNA, extracts hydrogen atoms from the deoxyribose backbone of DNA. From its position on the DNA, the calicheamicin radical is able to cause inter-strand crosslinks, reacting with molecular oxygen to produce DNA double-strand breaks [34]. The extent of damage is related to the arrest of cell cycle in G2/M and impacts DNA repair mechanisms and apoptotic pathways. Cells with impaired mechanisms of repair (e.g., ATM, poly (ADP-ribose) polymerase-1 enzyme (PARP-1)) [35] are particularly sensitive to the cytotoxic effects of GO. If damage is overwhelming, cells will die through the mitochondrial pathway of apoptosis predominantly utilized for GO-induced cell death. The calicheamicin-ϑII derivative triggers apoptosis in a p53-independent and death receptor/FADD-independent manner via activation of the mitochondrial permeability transition, cytochrome-*c* release, and activation of caspase 9 and caspase 3 [36]. Therefore, the mitochondrial protein Bcl-2 together with other anti-apoptotic BCL-2 family members reduce the cytotoxicity of calicheamicin-GO [37,38].

Not surprisingly from the above discussion, in vitro samples of pediatric AML showed 100,000-fold difference in calicheamicin sensitivity between the most sensitive and the most resistant patient samples [39]. The reasons for this variability and resistance are still not completely elucidated.

## 4. GO and Clinical Studies

Table 1 reports details of studies that used a combination of GO and chemotherapy that was initially withdrawn and then approved by the US FDA and subsequently in Europe by EMA. The first Phase I studies of GO, employing a dose of up to 9 mg/m^2^, was generally well-tolerated with neutropenia and persistent thrombocytopenia in 50% of patients, although dose-limiting toxicity and acute infusion-related clinical adverse events consisting of fever and chills were observed [40]. Subsequently, a Phase II trial in 142 older patients with first relapse of AML used GO at 9 mg/m^2^ (dose that stably saturates 75% of CD33) with an overall response rate (ORR) of 30% consisting of a complete remission (CR) rate of 16% and a CRp (without platelets recovery) of 13% [31]. Multivariate analysis revealed that increased CD33 expression was not related to response category or patient survival. Based on these results, GO was first approved by the FDA in 2000 [41]. Unfortunately, remissions were short-lived and at 12 months, less than 20% of patients remained in CR and 15% developed VOD or SOS, especially after allogeneic stem cell transplant [30]. In 2010, the registration Phase III trial SWOG S0106, required by FDA to actively maintain GO in the marketplace, disclosed disappointing results both in efficacy and toxicity. In fact, DA regimen (daunorubicin, 45 mg/m^2^ + cytarabine) + GO or DA alone (daunorubicin, 60 mg/m^2^) each showed similar results in terms of ORR (*p* = 0.36), CR (*p* =0.59), 5-year relapse free survival (RFS) (*p* = 0.40), and 5-year overall survival (OS) (*p* = 0.85) [42]. Fatal toxicity was 5% in the experimental arm vs. 1% in the standard arm. This was a lower fatality rate than expected with 3 + 7 and was the cause of the voluntary withdrawal of GO from the U.S. market, although other Phase 3 trials showed some benefits from the addition of GO.

In the open-label MRC AML15 trial, 1113 patients were randomly assigned to receive a single dose of GO (3 mg/m^2^) on day 1 of the induction course with 3 different induction schedules: DA; cytarabine (ARA-C), daunorubicin (DNR), and etoposide (ADE); or fludarabine, ARA-C, g-CSF, and idarubicin (FLAG-IDA). GO was associated with the same CR (83%) and 5-year survival (42%) rates, however, significant benefits of GO were seen in the group of favorable risk patients (overall survival 79% vs. 51%). CD33 expression classified as positive versus negative (+/−20% of blasts) was not predictive for response and there was no relationship with the degree of blast positivity [43]. The NCRI AML16 was conducted in 1115 older patients. The experimental group received GO 3 mg/m^2^ along with either DA or DNR/clofarabine. The study showed significant improvement in survival in patients receiving GO; the OS was 25% vs. 20% at 3 years (*p* < 0.05) [44]. No difference was seen in favorable-risk patients; however, it must be noted that the number of patients was limited.

In the GOELAMS-AML 2006 IR trial, 254 patients were treated with GO 6 mg/m^2^ and a standard 3 + 7 induction or a MidAc (mitoxantrone and intermediate doses of ARA-C) intensive consolidation course. Although differences were not significant, CR, OS, and event-free survival (EFS) rates were higher in the GO group than control group, however, early deaths were more frequent in the GO group [45,46].

Two different French studies used fractionated doses of GO before or in combination with induction chemotherapy in Mylofrance-1 [47] and Mylofrance-2 [48] studies, respectively. GO used at 3 mg/m^2^ saturated CD33 until its re-expression on new cells after 72 h, and for this reason showed less toxicity than at higher doses. This regimen was tested in the ALFA-0701 French trial in which 276 patients with de novo AML were randomized to receive standard 3 + 7 or 3 + 7 plus fractioned doses of GO [49]. EFS was significantly longer in the GO arm of low/intermediate risk and the interpretation generally given to these data was that the addition of GO does not change the rate of response after induction but improves EFS by reducing relapse [50]. Median RFS was 28 months in the GO arm and 11.4 months in the control arm. The EFS improvements are seen only in cytogenetic and ELN AML subgroups with a HR of 0.46, but are absent in the high-risk group [51].

From the safety point of view, VOD and hemorrhages increased with GO, although no differences in early mortality were observed. The relative advantages of low-toxicity doses were confirmed by the UK NCRI AML17 trial where a single dose of GO on day 1 at 3 mg/m^2^ vs. 6 mg/m^2^ were combined with ADE vs. DA. The 6 mg/m^2^ dose provided no advantage in response, disease-free survival, or OS compared with a 3 mg/m² dose; it was, however, associated with higher 30- and 60-day mortality [52]. The *NPM1* mutation has been associated with increase in CD33 expression, suggesting a possible benefit of GO in *NPM1*-positive AML. The results from a meta-analysis (MRC AML15, NCRI AML16, SWOG S0106, GOELAMS-AML2006 IR, and ALFA-070), however, showed no differential benefit [53]. Moreover, recent results of the prospective randomized AMLSG 09-09 Phase III study in *NPM1*-positive AML patients have not shown a difference in EFS from addition of GO to induction therapy with IDA, ARA-C, etoposide, and all-trans-retinoic acid, mainly for a higher early death rate in the GO arm [54]. Conversely, the addition of GO in the subgroup of *NPM1*-positive patients in the ALFA-0701 trial resulted in a significantly longer EFS [49]. In the same trial, low CD33 expression (<30% of positive blasts) seen in a minority of patients (13.7%) did not have an influence on the EFS benefit with GO. The same results were observed using a Cox proportional hazards model including treatment and CD33 expression as a continuous variable [50]. Nevertheless, a retrospective extensive mutational analysis of patients enrolled in this trial showed that higher CD33 expression levels in non-CBF blasts correlated with the presence of activating signaling mutations (*NPM1, FLT3, RAS*) [55]. Two Italian studies used GO in elderly patients: EORTC-GIMEM AML-17 trial is a randomized trial that evaluated GO (6 mg/m^2^ on days 1 and 15) followed by MICE standard chemotherapy (mitoxantrone, ARA-C, etoposide). The sequential combination of GO and MICE had no survival benefit but increased liver and hematologic toxicities [56]. The second Italian study was the EORTC-GIMEMA AML-19 trial where 237 patients aged >60 were randomized to receive either GO (6 mg/m^2^ on day 1 and 3 mg/m^2^ on day 8) or best supportive care. GO recipients experienced a longer median OS (4.9 vs. 3.6 months) and a higher 1-year OS rate (24.3% vs. 9.7%), with similar rates of adverse events (AE) [57]. In a randomized study using fractioned GO in 32 patients, the risk of VOD after HSCT was not increased, suggesting that GO does not induce excess post-transplant VOD/SOS or mortality and does not preclude the use of HSCT as consolidation treatment [58].

**Table 1 cancers-13-03214-t001:** DA: daunorubicin and cytarabine; ADE: Ara-C, daunorubicin, and etoposide; FLAG-Ida: fludarabine, cytarabine, granulocyte colony-stimulating factor, and idarubicin; MICE: mitoxantrone 7 mg/m^2^ D1, 3, 5; etoposide 100 mg/m^2^ D1–3; and cytarabine 100 mg/m^2^ D1–7; BSC: best supportive care.

Study	Treatment (Induction)	Patients	Median Age (Years)	ORR (CR + CRi)	OS	PFS/EFS	Early Mortality	Relapse	Comment	Reference
AAML0531	ADE (10 + 3 + 5) +/− GO 3 mg/m^2^ D + 6	1022	9.7 (0–30)	88% *v* 85%;	3 years: (69.4% *v* 65.4%; *p* = 0.39)	EFS (3 yrs): 53.1% *v* 46.9%; *p* = 0.04	6.6% *v* 4.1%; *p* = 0.09	3 yrs: 32.8% *v* 41.3%; *p* = 0.006	In pediatric patients, low doses of GO did not increase OS. EFS was improved with fewer relapses but slightly increased toxicity.	[59]
SWOG S0106	DA (dauno 45 mg/m^2^) + GO 6 mg/m^2^ D4 vs. DA (dauno 60 mg/m^2^)	595	47(18–60)	(69% v 70% *p* = 0.59 +76% v 74% *p* = 0.36	5 years: 46% v 50% *p* = 0.85	RFS (5 yrs) 43% v 42%; *p* = 0.4	5% v 1% *p* = 0.0062	5 yrs: 43% *v* 42%; *p* = 0.4	GO failed to show improvement in CR rate, DFS, or OS. Toxicity was significantly higher.	[42]
ALFA-0701	DA + GO 3 mg/m^2^ D 1,4,7 vs. DA	271	62.2	CR: 70.4% v 69.9%; CRp: 11.1% v 3.7%	Median 27.5 v 21.8 moths *p* = 0.16	Median EFS 17.3 v 9.5 months *p* = 0.0002. *No advantage in EFS with GO for poor cytogenetic risk*	6% v 4%	Median RFS 28.0 v 11.4 months	Fractionation of doses of the GO allows safe delivery of a much higher cumulative dose and improves outcomes.	[49,50]
MRC AML15	DA or FLAG-Ida +/− GO 3 mg/m^2^ D + 1	1113	49 (0–71)	CR: 82% v 83% *p* = 0.8; CRi: 3% v 4% *p* = 0.4	5 years: 43% v 41% *p* = 0.3 *OS improved with GO for good risk cytogenetics*	RFS (5 yrs) 39% v 35%; *p* = 0.09	11% v 10%	5-YRS 46% v 50% *p* = 0.12	A single low dose of GO associated with different induction schemes in young patients produced similar outcomes. However, a survival benefit for patients with favorable cytogenetics was evident.	[43]
NCRI AML16	DA (3 + 10) or DClofarabine +/−GO 3 mg/m^2^ D + 1	1115	67 (51–84)	CR: 62% v 58% *p* = 0.14; CRi: 9% v 10% *p* = 0.3	3 years: (25% *v* 20%; *p* = 0.05)	RFS (3 yrs) 21% v 16%; *p* = 0.04	9% v 8%	3-YRS 68% v 76% *p* = 0.007	Single low dose of GO in older pts. significantly reduced relapse risk, and improved OS with acceptable toxicity	[44]
NCRI AML17	ADE or DA + GO 3 mg/m^2^ v 6 mg/m^2^	788	50 (0–81)	CR: 82% v 76% *p* = 0.003; CRi: 7% v 10% *p* = 0.17	4 years: (50% *v* 47%; *p* = 0.3)	RFS (4 yrs) 44% v 38%; *p* = 0.3	3% v 7%; *p* = 0.02	4-YRS 46% v 54% *p* = 0.15	Single low dose of GO had similar disease-free and overall survival, but less toxicity with respect to intermediate dose.	[52]
GOELAMS AML 2006 IR	DA +/− GO 6 mg/m^2^	238	50 (18–60)	91.6% v 86.5% (*p* = NS)	3 years: 53% v 46%	EFS (3 yrs) 51% v 33%	10% v 4.5% (*p* = NS)	/	In patients with intermediate cytogenetics AML, GO failed to improve OS	[45,46]
EORTC-GIMEMA AML-17	MICE +/− GO 6 mg/m^2^ D 1, 15	472	67 (60–75)	CR: 39% v 41%; CRp: 9% v 8%	2.5 years:16% v 21.7% *p* = 0.07	EFS (1 yr) 18%	17% v 12%	/	Combining two upfront doses of GO 6 mg/m^2^ with sequential chemotherapy does not benefit older patients with AML, is too toxic for those >70 years.	[56]
EORTC-GIMEMA AML-19	GO (6 mg/m^2^ on D1 and 3 mg/m^2^ on D8) vs. BSC	237	77 (62–88)	CR: 8.1% CRi: 16.2%	1 year: 24.3% v 9.7%	Median DFS was 5.3 months	7%	/	Older patients treated in first line with GO showed significantly improved OS in all subgroups, with comparable toxicity than BSC.	[57]

## 5. GO and Current Indications

In 2017, FDA granted approval of GO with 2 indications: (1) treatment of newly diagnosed CD33-positive AML in adults in combination with DNR and ARA-C at 3 mg/m^2^ (up to 5 mg) on days 1, 4, and 7 or as single-agent regimen at 6 mg/m^2^ on day 1 and 3 mg/m^2^ on Day 8; and (2) treatment of relapsed or refractory CD33-positive AML in adults and in pediatric patients >2 years at 3 mg/m^2^ on days 1, 4, and 7 [60,61]. In 2018, EMA approved GO for patients >15 years with previously untreated, de novo CD33-positive AML, at 3 mg/m^2^ (up to 5 mg) on days 1, 4, and 7 + DNR 60 mg/m^2^/day on days 1 to 3, and Ara-C 200 mg/m^2^/day by continuous infusion on days 1 to 7 [62,63].

## 6. In Vitro Relationship between CD33 Expression and GO Efficacy

An unambiguous relationship between CD33 expression and GO efficacy has been clearly demonstrated in in vitro studies. Specifically, GO rapidly and specifically targets CD33+ cells, followed by its internalization and subsequent cell death. Van der Valden et al. first showed that calicheamicin, conjugated to a CD33 antibody, was able to induce apoptosis in vitro in CD33-expressing cells but was unable to induce apoptosis in CD33-negative cells (i.e., lymphocytes) [25]. Furthermore, GO-induced cytotoxicity was strictly correlated with cell surface expression of CD33; specifically, higher CD33 expression levels were associated with a rise of GO binding to CD33 antigenic sites and thus accelerated clearance of AML blasts [3,64,65]. Walter et al. showed that not only the level of CD33 expression, but also the rate of endocytosis, induced the extent of GO-induced cytotoxicity. By manipulating the endocytic process, they found that a faster internalization of antibody-bound CD33 could theoretically lead to enhanced GO-induced cytotoxicity. They also showed that disruption of ITIMs by point mutations not only avoided the effective internalization of antibody-bound CD33, but also significantly decreased GO-induced cytotoxicity [64].

Good responders among GO recipients also expressed higher mean CD33 expression levels that were inversely correlated with the expression of the low ATP-binding cassette subfamily B-member 1 (ABCB1) mediating drug efflux [3]. Using an in vitro model, Jawad et al. found that GO could induce 34% reduction in CD34+CD38−CD123+ leukemic stem and progenitor cells (LSPC), whereas normal CD34+CD38− hematopoietic stem cells were insensitive to this agent. Specifically, LSPCs that overexpressed CD33 (*p* = 0.01) were *p*-glycoprotein-negative (*p* = 0.008) and responded better to GO with an internal tandem duplication (ITD) of the *FLT3* gene (*p* = 0.006) [65]. In summary, the majority of in vitro studies using pulse labeling with GO showed a continuous renewed membrane expression of CD33 antigens, which could significantly increase the internalization process and thereby the intracellular accumulation of the drug. Taken together, these data suggest novel therapeutic approaches for improvement of clinical outcome of patients treated with GO.

## 7. In Vivo Relationship between CD33 Expression and GO Efficacy

Contradictory in vivo results have been reported on the relationship between CD33 expression and GO efficacy. No interaction was found when CD33 expression was assessed as a continuous co-variable [31,49] or using a 20% cut-off [43]. However, higher response rates were observed among patients with CD33 expression of ≥98% in a Phase 2 trial [66] or when CD33 expression was evaluated as mean fluorescence intensity (MFI) using an isotype antibody as control [3].

In AAML0531 trial, 825 pediatric patients were randomly assigned to one of two study arms: a backbone of standard chemotherapy alone (No-GO arm) or in combination with 3 mg/m^2^ GO administered on day 6 of induction I and day 7 of intensification II (GO arm). The study population was divided into four quartiles based on CD33 expression. The median MFI for quartile (Q) 1 to 4 was as follows: Q1, 34.61 (range, 2.68–67.00, n = 208); Q2, 100.7 (range, 67.13–146.94, n = 205); Q3, 207.01 (range, 147.00–296.38, n = 206); and Q4, 435.9 (range, 296.98–1351.00, n = 2060). The addition of GO to standard chemotherapy resulted in an improvement of EFS in Q2–Q4 patients with high CD33 expression (GO vs. no-GO, 5-year EFS: 53% vs. 41%, *p* = 0.005), whereas patients with low CD33 expression (Q1) did not benefit from GO (GO vs. no-GO, 5-year EFS: 53% vs. 58%, *p* = 0.456). In all risk groups, patients with low CD33 expression had similar outcomes regardless of GO exposure, whereas the addition of GO to conventional chemotherapy caused a significant decrease in relapse and disease-free survival (DFS) rates for patients with higher CD33 expression [67].

In adults UK-NCRI-AML17 trial, no effect of GO on survival in non-CBF patients was observed in the different quartiles, except in Q4 where addiction of GO significantly reduced the risk of relapse [18].

In a post-hoc analysis conducted on 200 adult patients belonging to the ALFA-0701 study, which evaluated CD33 expression as a binary variable defined by a 70% cutoff, GO was associated with an improvement of EFS and RFS rates in patients with high CD33 expression even after adjustment for cytogenetics and *NPM1/FLT3-ITD* mutations [17]. If these results are confirmed in future retrospective or prospective trials, including quantification of both CD33 percentage and CD33 MFI, this 70% expression cut-off can be used as a practical biomarker for GO efficacy. A recent study assessed the association between signaling pathway mutations and benefit of GO, comparing the presence of mutations with CD33 expression levels on AML blasts [55]. The benefit of GO on EFS was correlated with CD33 expression levels among the different gene mutations, with high levels detected on signaling mutation-positive AML blasts. Among patients harboring epigenetic mutations, CD33 expression was significantly higher in patients with signaling pathway mutations (98% vs. 60%; *p* < 0.001); these features were not identified in patients with *NPM1* or spliceosome mutations [55]. Therefore, the authors concluded that the benefit of GO was primarily observed in patients harboring signaling pathway mutations associated with high CD33 expression.

## 8. Prognostic Impact of Cytogenetic Alterations and Molecular Profile on GO Efficacy

A large meta-analysis showed that the addition of GO to chemotherapy positively correlated with significantly longer survival in patients with good and intermediate cytogenetic profiles [53]. Similar survival rates were observed in those presenting lower expression of CD33 [53]. Core binding factor (CBF)-AML was inversely correlated with CD33 expression [16,18] and the authors speculated that the initial event, t(8;21)/inv(16)/t(16.16), occurred very early in pre-leukemic CD33-negative cells [68]. However, additional driver mutations (i.e., tyrosine kinase mutations) causing proliferative AML clones may arise when the blasts phenotypically express CD33 [69]; therefore, GO may eliminate the proliferative blast cells and spare CD33-negative pre-leukemic cells. In addition, the high response rate to GO in CBF-AML may be associated with a high sensitivity of CBF-AML blasts to calicheamicin [70].

11q23/lysine methyltransferase 2A (KMT2A) rearrangements represent recurrent cytogenetic aberrations in AML, often detected in pediatric AML and associated with high CD33 expression on leukemic blasts [71]. In the COG AAML0531 trial, 215 patients presented 11q23/KMT2A rearrangements. Among this group, patients who received GO plus chemotherapy showed a significantly better EFS compared to those treated with chemotherapy alone (5-year EFS: 48% vs. 28%, *p* = 0.002). It should be noted that the 5-year OS in this study was not significantly different between the two treatment arms (5-year OS: 64% vs. 53%, *p* = 0.053) [72].

In APL patients, blasts express the CD33 antigen in nearly 100% of cases affording the opportunity to use GO in this setting. In vitro reports have demonstrated the efficacy of GO in all-trans retinoic (ATRA)- or arsenic trioxide (ATO)-resistant APL cells which translated into second complete remission in clinical trials [73,74]. ATRA plus ATO with or without GO is associated with a safe toxicity profile and a high rate of remission (CR rate 90% and 81%, respectively) in high-risk APL patients [75]. Recently, a Phase II trial (SWOG S0535) evaluated the efficacy of ATRA and ATO combination with GO in high-risk APL patients. Complete remission was observed in 86% of the 70 evaluable cases and the 3-year OS and RFS were 86% and 91%, respectively [76]. Therefore, this chemotherapy-free approach might be curative in high-risk patients and, in the future, might be considered the gold standard of care in this setting.

The use of GO has shown very promising results in patients harboring the *NPM1* or *FLT3* mutations. *NPM1* alterations have been observed in 25% to 35% of AML patients and, in most cases, are associated with a normal cytogenetic profile (45–60%) [77]. *NPM1* mutations were found to be significantly increased in prevalence with increasing CD33 expression [13,16]. In the subgroup of *NPM1*-positive patients of ALFA-0701 trial, the addition of GO produced a positive effect on EFS, but not on OS [49]. A recent Phase III trial (AMLSG 09-09) assessed the potential efficacy of GO in induction (3 mg/m^2^ on day 1) and consolidation courses (3 mg/m^2^ on day 1 of the first consolidation cycle) in *NPM1*-mutated patients eligible for intensive chemotherapy [54]. The addiction of GO did not affect the 2-year EFS (*p* = 0.10). Among patients who achieved a CR or CR with incomplete hematologic recovery, GO significantly reduced the risk of relapse (*p* = 0.005). Interestingly, the use of GO improved the 2-year EFS of *FLT3* wild-type cases, but not that of the *FLT3-ITD* mutated patients (*p* = 0.002).

*FLT3-ITD* mutations are present in approximately 20% of AML patients, and correlate with high expression of CD33 antigen on leukemic blasts [77]. The use of GO in combination with standard chemotherapy has shown increased OS, RFS, and EFS rates in AML patients harboring *FLT3-ITD* mutations [49,78]. In a retrospective analysis of the COG AAML03P1 and AAML0531 trials including the *FLT3-ITD* mutated patients, the addition of GO to chemotherapy determined a lower relapse rate (37% vs. 59%, *p* = 0.02) compared with chemotherapy alone [79]. In the group of patients receiving HSCT in the first complete remission, previous exposure to GO correlated with a significant reduction of the relapse rate (22% vs. 56%, *p* = 0.003). Patients who presented a high *FLT3-ITD* allelic ratio (>0.4) at baseline showed a lower risk of relapse when GO was administered prior to HSCT (15% vs. 53%, *p* = 0.007). Biological predictors of response to GO are summarized in Table 2.

## 9. Relationship between CD33 Single Nucleotide Polymorphisms and GO Efficacy

Several reports have assessed the potential relationship between CD33 antigen genotype and clinical response to GO. A retrospective study (AML02 trial) found that a nonsynonymous coding change (C > T; Ala14Val) in the splice enhancer region of the CD33 gene exon 2 was significantly associated with positive response to GO (*p* = 0.02), whereas CD33 transcript and protein expression were not (*p* > 0.2) [27]. More recently, Mortland et al. genotyped four CD33 single-nucleotide polymorphisms (SNPs): rs35112940 (G > A; Arg304Gly), rs12459419 (C > T; Ala14Val), rs2455069 (A > G; Arg69Gly), and rs1803254 (G > C; 3′UTR) in pediatric patients undergoing induction chemotherapy with or without GO (COG-AAML03P1 trial; n = 242 and St. Jude AML02 trial; n = 172). Patients homozygous for the rs12459419 variant allele (TT) were more likely to have a favorable risk outcome than CC and CT genotypes (52% vs. 31%, *p* = 0.034) and significantly lower CD33-positive blasts than other genotypes (*p* < 0.001) [80]. Among the 816 patients (aged 0 to 29 years) included in the AAML0531 trial who were genotyped for the SNP rs12459419, 51%, 39%, and 10% of the patients expressed the CC, CT, TT genotype, respectively. In this cohort, GO addition provided several advantages, including decreased risk of relapse and RFS only in cases with CC genotype [81]. A recent analysis from the MRC AML15 and NCRI AML17 trials (younger adults with AML 13–69 years) demonstrated a similar distribution of CC, CT, and TT genotypes (47%, 44%, 9%, respectively), however, OS and RFS were not influenced by GO in the other genotype subsets [82]. Additional CD33 SNPs genotyping studies focused on five further SNPs potentially affecting GO efficacy in adult AML patients—rs1803254(G > C; 30 UTR), rs35112940(G > A; Arg304Gly), rs2455069(A > G; Arg69Gly), rs61736475(Ser305Pro), and rs201074739 (CCGG deletion) [83]. Patients with rs1803254 GG (*p* = 0.009), rs35112940 GG (*p* < 0.001), rs2455069 GG (*p* = 0.005), rs61736475 TT (*p* = 0.002), and rs201074739 CCGG/CCGG (*p* = 0.002) genotypes all showed a lower relapse rate after receiving GO.

A composite CD33 pharmacogenetics score (CD33_PGx6_score) using six CD33 SNPs (rs12459419, rs2455069, rs201074739, rs35112940, rs61736475, and rs1803254) has recently been proposed to evaluate the interaction between CD33 expression and GO efficacy in 938 de novo AML patients (aged 0–29 years). Patients with a CD33_PGx6_score of 0 or higher showed superior CD33 expression levels compared to patients with a score of less than 0 (*p* < 0.001). In addition, patients with a score of 0 or higher showed a better DFS in the GO versus non-GO arms (62.5% ± 7.8% vs. 46.8% ± 8.3%, respectively; *p* = 0.008) and a lower risk of relapse (28.3% ± 7.2% vs. 49.9% ± 8.4%, respectively; *p*< 0.001). No improvement from GO was observed in patients with a CD33-PGx6_score of less than 0 [83]. If further validated, these findings hold promise to guide efficient use of GO in patients with AML.

## 10. GO Resistance

Multidrug resistance (MDR) is a frequent event, resulting in the development of cross-resistance to several cytotoxic drugs. P-gp, which plays a crucial role in MDR, is a membrane glycoprotein that prevents cytotoxic agent internalization into cells and reduces intracellular drug accumulation [84,85]. In vitro evidence indicates that GO has a more cytocidal effect on cell lines that overexpress P-gp, even if these cells have substantial levels of CD33 on the cell surface [84]. The occurrence of this resistance mechanism has been shown by the concomitant in vitro use of GO and MDR modifiers (i.e., PSC833 and MS209) in resistant cell sublines [86,87] suggesting that the combination of GO and MDR modifiers might potentially be considered as an ideal therapeutic approach for P-gp-expressing leukemic cells.

Other resistance mechanisms have also been reported: the role of the anti-apoptotic proteins bcl-2 and bcl-x in promoting resistance to GO therapy has been described [87,88]. Overexpression of bcl-2 and bcl-x decreased the impact of GO, however, GO’s effect was increased by the bcl-2 antisense oligonucleotide. Resistance to GO is mediated by Bax, Bak, and stress-activated protein kinase. GO induced pro-apoptotic activation of Bak, Bax, and stress-activated protein kinase in responsive AML cells but not in resistant cells [89]. By inhibiting P-gp and inducing mitochondrial apoptosis, the peripheral benzodiazepine receptor ligand PK11195 increased the sensitivity of AML cells to standard chemotherapeutics [90]. It also made AML cells more responsive to GO.

Activation of survival signaling pathways, including PI3K/AKT, MEK/ERK, and JAK/STAT, has been linked to GO resistance in AML cells in vitro [91]. MK-2206, an AKT inhibitor, restored GO and calicheamicin resistance in resistant AML cells, suggesting the possibility that delivering GO to bone marrow is crucial for maximizing GO impact. The effect of GO was reduced by an excess of circulating CD33-positive cells, resulting in worse outcomes [25]. High blast cell counts, on the other hand, are an unfavorable prognostic factor in leukemia treated with other anti-leukemic drugs.

Many in vitro agents other than MDR modifiers increase GO sensitivity. G-CSF amplified the effect of GO, causing AML cells to reach the G2/M and hypodiploid phases [92] and valproic acid, a histone deacetylase inhibitor, also increases GO efficacy [91]. In clinical trials, however, the synergistic impact of GO with these agents has not been elucidated.

Several groups have proposed other resistance mechanisms, such as alternative GO pharmacokinetics and the reduction of CD33 on leukemia cells [22,87,93]. It is likely that multiple mechanisms play a role in GO resistance.

## 11. GO as Maintenance Therapy in AML

Several studies have investigated the potential role of GO as maintenance therapy in AML. In a phase 3 trial, AML patients >60 years of age who had achieved a CR1, were randomized. This was between three cycles of GO (6 mg/m^2^ every 4 weeks) or no post remission therapy to assess whether GO improved outcomes. There were no significant differences between both treatment groups with regard to relapse probabilities, non-relapse mortality, OS, or DFS (17% vs. 16% at 5 years) [94]. The SWOG S0106 trial investigated the role of GO in combination with induction and as single agent in maintenance therapy (three doses of GO 5 mg/m2 every 28 days) in younger patients. The disease-free survival (DFS) was not significantly better in the GO group (*p* = 0.97) and was not superior to the observation group in any cytogenetic risk group [42].

According to these studies, GO seemed to be ineffective as maintenance in all AML age groups and its use as maintenance has not been investigated further.

## 12. CD33 Bispecific Antibodies and CAR-T CD33

Following the success of blinatumomab, a CD19-targeted bispecific antibody for the treatment of acute lymphoblastic leukemia, clinical trials have investigated CD33-targeted bispecific antibodies for AML. Bispecific T-cell engagers (BiTE^®^) antibody constructs are generated combining the scFv domains of two separate antibodies on a single polypeptide chain [95]. One scFv domain is designed to identify an epitope on CD3 and the other is designed to bind a tumor-associated antigen (TAA) expressed by tumor cells (e.g., CD33 in the case of AML). The two scFv domains are connected by a short exible glycine linker, which allows the two domains to bend and/or twist against each other. T-cell activation is also caused by BiTE^®^-induced CD3 clustering on the cell surface [96], as evidenced by increased expression of the activation markers CD69 and CD25.

AMG330 is a short-acting BiTE^®^ canonical molecule that is administered in a 2–4-week period as a continuous intravenous (IV) infusion. Preclinical studies have shown that the AMG330′s anti-CD33 x anti-CD3 construct is cytotoxic even when target cells have low CD33 antigen density, making it a candidate to target a wide variety of CD33-positive leukemia including AML. Furthermore, AMG330 efficiently recruits and activates residual T cells to primary AML cells in patient samples in an in vitro, long-term, co-culture system [97]. In a Phase 1 trial (NCT02520427), 55 patients with relapsed or refractory (r/r) AML were enrolled to evaluate the safety and maximum tolerated dose of AMG330 [98]. Eight of the forty-two evaluable patients (19%) responded to AMG330; in particular, three obtained complete responses (CR), four showed incomplete hematologic recovery (CRi), and one exhibited a morphologic leukemia-free condition (MLFS).

AMV564 is a tetravalent anti-CD33x anti-CD3 tandem diabody (TandAb) construct given as a 14-day on/14-day off continuous IV infusion. Since AMV564 is tetravalent (two CD3 binding sites and two CD33 binding sites), it has a higher affinity for both targets. AMV564 has a longer half-life than monovalent bispecific antibodies due to its higher molecular weight. AMV564 is currently testing in a Phase 1 trial (NCT03144245) to determine its safety and efficacy in treating patients with r/r AML [99]. Thirty-six patients have enrolled in this study to date. Three of the thirty-five evaluable patients (9%) had an objective response, with one CR, one CRi, and one partial response (PR). Seventeen of the thirty-five evaluable patients (49%) showed blast reduction.

Remarkable results were achieved with CAR-modified T-cells targeting CD19 in relapsed/refractory patients with diffuse large B-cell lymphoma (DLBCL) [100] and pediatric B-cell acute lymphoblastic leukemia (ALL) [101] (50–90% CR), which inspired development of CAR-T cellular therapy for other indications, such as AML. Identifying a myeloid target for CAR-T cellular therapy has proven challenging, since surface antigens are shared by malignant myelogenous cells and normal hematopoietic stem cells, potentially resulting in prolonged myelotoxicity when tested in clinical trials. A Phase I clinical trial (NCT03126864) investigated the feasibility and safety of autologous T-cells, modified to express a CD33-targeted CAR with 4-1BB and CD3ζ endo-domains and co-expressed with truncated human epidermal growth factor receptor (HER1t), in patients with r/r AML. Ten adults with r/y AML were enrolled. Patients had received a median of 5 (range 3–8) prior treatment regimens; 3 underwent prior HSCT. Apheresis was collected for 8 patients; 4 had CD33-CAR-T cells produced which met release pre-specified release criteria for infusion. Three patients received CD33-CAR-T cells at the first dose level (0.3 × 106 CD33-CAR-T/kg) and one patient died before receiving their cells. Unfortunately, all three patients who received CD33-CAR-T cells have died, due to disease progression [102]. A single case report of a patient with relapsed AML treated with CD33-CAR-T cells suggested therapeutic activity with associated symptoms and increase in inflammatory cytokine levels (NCT01864902) [103]. These trials demonstrate challenges associated with CAR-T therapy for patients with r/r AML, which relate to manufacturing using viral vectors, rapid disease progression and risk for infection, and coordination of administration of this therapeutic modality. Clinical trials are ongoing to test CAR-T cells targeting antigens other than CD33 on myeloid blasts, including CD123 (NCT03766126, NCT03114670, NCT03190278, NCT04230265, NCT03631576), IL1 receptor accessory protein (IL1RAP) (NCT04169022), CLL-1 (NCT04219163), CD38 (NCT04351022), and FLT-3 (NCT03904069).

In Table 3 we summarize the most important resistance mechanisms to GO, GO related adverse events in ALFA-0701 trial, and clinical trials with CD33 bispecific antibodies.

## 13. Conclusions

AML is a paradigm for the therapeutic use of monoclonal antibodies because malignant cells are readily accessible and express well-defined cell surface antigens. Approximately, 85–90% of adult and pediatric AML cases are CD33-positive on more than 20–25% of the leukemic blasts, as defined by the expression of this specific antigen. Most efforts have focused on exploiting CD33 as a target in this disease, and several clinical trials have confirmed the anti-leukemic activity of GO in CD33-positive AML cells and have demonstrated improved outcomes in AML patients.

In vitro studies have clearly demonstrated a relationship between CD33 expression and GO efficacy. GO is rapidly and specifically targeted to CD33+ cells, followed by its internalization and subsequent induction of cell death. In vivo studies have shown the benefit of GO on outcomes in patients with high CD33 expression, evaluated as a binary variable defined by a 70% cutoff, even after adjustment for cytogenetics and *NPM1/FLT3-ITD* mutations. Furthermore, the benefit of GO is primarily observed in patients harboring signaling pathway mutations associated with high CD33 expression. Several reports have also assessed the potential relationship between CD33 antigen genotype and the clinical response to GO. The CD33_PGx6_score based on the genotyping of six different *CD33* SNPs could in the future be a valid diagnostic method to identify potential good responders to GO therapy.

Over the years, flow cytometry, cytogenetics, molecular techniques, and genotyping studies of CD33 SNPs have provided a comprehensive analysis of promising biomarkers for GO responses and have potentially helped to identify subgroups of patients that may benefit from GO addition to standard chemotherapies. Increased understanding of molecular mutations, altered intracellular pathways, and their potential relationship with CD33 expression may open new therapeutic landscapes based on combinatorial regimens in an AML scenario. For these reasons, ongoing studies are assessing the efficacy of GO in combination with other target therapies, such as *FLT3-ITD* inhibitors (NCT03900949, NCT04385290, NCT04293562) and Bcl-2 inhibitors (NCT04070768, NCT04070768). In addition, clinical trials investigating CD33-targeted bispecific antibodies for AML are in progress with promising results in r/r AML patients.

## Figures and Tables

**Figure 1 cancers-13-03214-f001:**
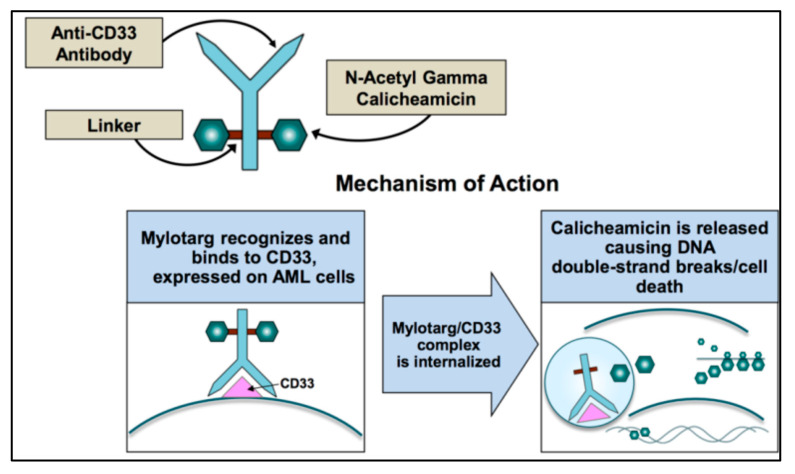
Mechanism of action of gemtuzumab ozogamicin.

**Table 2 cancers-13-03214-t002:** Predictors of response to GO.

Predictors of response to GO	Favorable cytogenetic risk Intermediate cytogenetic risk CBF mutated AML [t(8;21)/inv(16)/t(16.16)] 11q23/KMT2A rearrangements *NPM1/FLT3-ITD* mutated AML Signaling mutation-positive AML CD33_PGx6_score of 0 or higher de novo or relapsed/resistant APL
Uncertain predictive role on response to GO	CD33 expression level Multidrug resistance (MDR) mediated by ATP-dependent drug transporters such as P-glycoprotein Sensitivity to calicheamicin in vitro CD33 single-nucleotide polymorphism rs12459419 genotype
Unfavorable predictors of response to GO	Adverse cytogenetic risk CD33_PGx6_score less than 0 Overexpression of bcl-2 and bcl-x PI3K/AKT, MEK/ERK, and JAK/STAT in vitro activation

**Table 3 cancers-13-03214-t003:** The most important resistance mechanisms to GO, GO related adverse events in ALFA-0701 trial, and clinical trials with CD33 bispecific antibodies.

Resistance mechanisms to GO	Overexpression of P-glycoprotein which favors multidrug resistance (MDR) mechanisms. Overexpression of the anti-apoptotic proteins bcl-2 and bcl-x. Activation of survival signaling pathways including PI3K/AKT, MEK/ERK, and JAK/STAT. Overexpression of multidrug resistant-related protein 1 (MRP1) on AML cells.
Most important GO-related adverse events in ALFA-0701 study [49,50]	In the ALFA-0701 study, the most frequent (≥1%) adverse reactions that led to permanent discontinuation in the combination therapy study were thrombocytopenia, VOD, hemorrhage, and infection [49,50]. VOD was reported in six (4.6%) patients during or following treatment, and two (1.5%) of these reactions were fatal. Thrombocytopenia with platelet counts <50,000/mm^3^ persisting 45 days after the start of therapy for responding patients (CR and incomplete platelet recovery) occurred in 22 (20.4%) patients.
CD33 bispecific antibodies in clinical trials	AMG330 Phase 1 trial (NCT02520427); 55 patients enrolled [98]. Response: 4 CR, 3 CRi, 1 MLFS AMV564 Phase 1 trial (NCT03144245); 36 patients enrolled [99]. Response: 1 CR, 1 CRi, 1 PR, 17 of 35 patients had blasts reduction AMG673 Phase 1 trial (NCT03224819); 30 patients enrolled [104]. Response: 1 CRi, 11 of 27 patients had blasts reduction

## Data Availability

The raw data supporting our findings can be requested from the corresponding author.

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
