# Peer review of "CD33 Expression and Gentuzumab Ozogamicin in Acute Myeloid Leukemia: Two Sides of the Same Coin"

_cancers, 2021, doi:10.3390/cancers13133214_

Round 1

Reviewer 1 Report

This is a comprehensive review on the role of CD33 in both a diagnostic and therapeutic perspective. The review is well written and contains no obvious misinformation. Thsi reviewer has minor comments only:

  • The paper is somewhat "long" to read. The review would e.g. benefit from a Figure demonstrating mode of action of CD33 treatment.
  • Recent approaches to target CD33 by CAR-T  technology should be discussed.
  • Maintenance issues with anti-CD33 treatment should be included.

Author Response

1) a figure including the mechanism of action of GO has been added

2) in the paragraph 12 has been added a part on CD33 CAR-T

3) a new paragtaph (number 11) has been added on GO as maintainance therapy

Reviewer 2 Report

Comments for the authors:

Monoclonal anti-CD33 gemtuzumab ozogamicin (GO) is the first targeted therapy approved for the treatment of acute myeloid leukemia (AML), but it was voluntarily withdrawn from the market in 2010 due to its potential side effects. After 7 years, GO was re-approved based on new data, including insights into its mechanism of action on its target receptor CD33, expressed on myeloid cells. This manuscript provides an overview of the current biological information and clinical data from several studies investigating GO. Overall, the manuscript is well-written and informative; however, for a better understanding of this topic by a broad readership, I suggest that the authors provide either a table or a simplified figure to represent the study findings concisely. Several minor issues have been noted, which should be addressed by the authors before the manuscript is accepted for publication. Some suggestions are provided below:

  1. Please prepare a table or a simplified illustrative diagram to depict the pathogenesis of CD33 in AML and the mechanism of action of GO in the treatment of AML. Also, could you provide a simplified illustrative diagram to depict CD33 domains and its functions (e.g. transmembrane, dimer, IgV, etc.) and how are they different from other isoforms?
  2. For a better understanding of this topic by the readers, the authors should provide a table listing and summarizing the relationship between cytogenetic alterations, molecular profiles, CD33 SNP, and GO efficacy in the eighth and ninth sections of this manuscript.
  3. Finally, please provide an overview table summarizing the current GO susceptibility/resistance, side effects, and the application of CD33 bispecific antibodies. In addition, the combined treatment should also be considered in the summary.

Author Response

1) a figure including GO mechanism of action has been added

2) a table listing and summarizing the relationship between cytogenetic alterations, molecular profiles, CD33 SNP, and GO efficacy in the eighth and ninth sections of this manuscript has been added

3) an overview table summarizing the current GO susceptibility/resistance, side effects, and the application of CD33 bispecific antibodies has been added